



# Impact of natural parameters on rock glacier development and conservation in subtropical mountain ranges. Northern sector of the Argentine Central Andes

Ana P. Forte[1], Cristian D. Villarroel[2] and María Y. Esper Angillieri[1]

[1] CONICET – CIGEOBIO: Geosphere and Biosphere Research Center. Faculty of Exact, Physical and Natural Sciences, National University of San Juan, San Juan, 5400, Argentina
[2] CONICET - Department of Geology. Faculty of Exact, Physical and Natural Sciences, National University of San Juan, San Juan, 5400, Argentina

*Correspondence to*: Ana P. Forte (anapau.forte@gmail.com)

**Abstract.** This paper presents a detailed rock glacier inventory used in determining how the various natural parameters affect a mountain periglacial environment. This study was undertaken in a northernmost sector of the central Argentine Andes, in an area stretching between 31°02′ and 31°22′ S latitude. This is a high and arid subtropical region where permafrost and cryogenic processes are predominant, featuring as well as a large number of rock glaciers and associated periglacial landforms. Rock glaciers inventory was based on geomorphological characterization with optical remote sensing

data and field description information. The study region covers 630 km², with 3,25 % of this area showing 402 rock glaciers and protalus rampart features. In total, 172 rock glaciers have been identified, 48 of which are considered active. In such a sector, the protalus rampart range shows the largest landform occurrences, though fossil and inactive rock glaciers are usually larger and are developed over a larger attitudinal distribution. Based on previous studies, we have considered that the study of active rock glaciers is an effective approach to assess the current state of periglacial environment evolution.

Therefore, we analysed their spatial distribution and their relationship through different variables, by resorting to statistical analysis and a frequency ratio method. The chosen natural parameters were: Lithology, Elevation, Latitude and Longitude, Aspect, Slope and Annual Potential Solar Incoming Radiation. Analytical results have demonstrated that active rock glaciers landforms lie above 3.600 m.a.s.l. elevations, on 11° to 28° slopes with predominating south facing aspect and relatively low solar radiation. The statistical analysis shows that elevation, lithology and aspect are the most influencing factors for current

periglacial environment development while rock glacier conservation is mainly controlled by lithology. On the study area, the annual potential solar radiation show high values but there is not any significant difference between landform and, therefore, it is scarcely influential. The research is carried out over a high mountain area where poor accessibility hindered the chances for obtaining systematic data on weather and environment. A simple and low coast methodology was used to analyse an area where no studies on rock glacier distribution had been made before. This information gains special

importance because Argentina has recently instituted a national law for glacial and periglacial environment protection and



conservation. Therefore, this research and its results may contribute a significant step toward knowing the number, features and distribution of rock glacier bodies lying in a scarcely studied region.

## 1 Introduction

In arid mountain ranges, such as those of the Argentine Central Andes (Lliboutry, 1999), the incorporation of cryospheric science in water resource research has gained a special significance for proper water resources planning and management, as well as for reducing the vulnerability upon the regional ecology and human activities. Periglacial environment (Tricart, 1968; French, 2007) is particularly frequent and well developed in dry and continental mountain areas (Brenning and Trombotto, 2006). Their distribution has been studied by several authors (Brenning et al., 2007; Johnson et al., 2007; Esper Angillieri, 2010). The impact of climatic factors upon a periglacial environment, mainly temperature and precipitation, has been studied by Brazier et al. (1998); Humlum (1998); Brenning (2005); Brenning and Trombotto (2006) and Azócar et al. (2016). Periglacial environments, however, are controlled not only by climatic factors, especially in mountain environments (French, 2007) but by other influential parameters. Influence of regional lithology (Evin, 1987; Humlum, 2000; Ikeda and Matsuoka, 2006), potential solar radiation and aspect (Schrott H., 1991; Funk and Hoelzle 1992; Etzelmüeller et al., 2001; Boehner, 2009), elevation and local topography must be considered as having the same importance as temperature and precipitation.

Rock glaciers occurrences are direct indicators of mountain permafrost (Barsch D., 1996, Haeberli et al., 2006; Azócar et al., 2016), and could be considered as a proxy of current periglacial environment state. Unlike glaciers, rock glaciers show a delay in response to climate change and are conditioned by different parameters which control the velocity and degree of development, preservation and/or deterioration. Therefore, rock glaciers distribution faithfully reflect the past and present mountain permafrost conditions. Based on their dynamic, rock glaciers are classified as active, inactive and fossil rock glaciers (Wahrhaftig and Cox, 1959). Besides, the surface speed of rock glaciers is a factor closely related to their equilibrium state with the surrounding environment. Since past and present climatic patterns and environmental factors may impact upon current rock glacier distribution, the present study has adopted the above classical classification so as to understand coherently the way such factors control rock glacier distribution. Active and inactive landforms could be considered as an entire category called intact rock glacier (Barsch, 1996). Active rock glaciers preserve and increase their ice content and could be considered as a water reserve (Arenson and Jakob, 2010 and Azócar and Brenning, 2010). Whereas inactive rock glaciers, due to their unbalance with the environment, keep losing ice and are considered water sources. Finally, relict or fossil rock glaciers have lost all their ice content and indicate unfavourable actual periglacial environmental conditions, although some periglacial processes could still exist in a close proximity. Protalus ramparts have been considered an expression of mountain permafrost creep and, as such, they may be considered as embryogenic rock glaciers (Barsch, 1996).



In the past decade, a number of rock glaciers inventory survey and associated research campaigns have been conducted along the Argentine Central Andes, e.g. Trombotto et al. (2012); Villarroel (2013) and Tapia Baldis and Trombotto (2015). Rock glaciers distribution analyses were made by Brenning and Trombotto (2006); Esper Angillieri (2008); Trombotto and Borzotta (2009); Schreiber (2015) and Forte et al. (2016). In addition, Arenson et al. (2010) studied two rock glaciers on Pachon River basin and Forte et al. (2013) conducted a geophysical research in two rock glaciers on Frío river basin. Villarroel et al. (2016) used geophysical methods over five rock glaciers in San Juan river basin, and Tapia Baldis et al. (2016) presented a preliminary MAAT model used for the SW mountain ranges of San Juan.

This paper is aimed at analyzing the relationship between spatial rock glacier distribution and the impact of climatic, geologist, topographic and morphometric parameters, by using a frequency ratio method for the statistical analysis. Also presented is a rock glacier inventory of a Northern sector of the Argentine Central Andes, specifically, the watershed areas of La Salina river, included Frío, Los Bagres and De Las Salinas river basin. This information is essential for conducting regional studies on periglacial environment distribution, especially in high mountain areas where poor accessibility reduces the chances of obtaining climatic data, while it steeply increases the costs of systematic field studies. In Argentina, a recent National Law for protecting glacial and periglacial environment has been put in force, which means that this kind of research has a significant regional importance. The results presented in this work will therefore constitute an important progress for knowing the number, features and distribution of rock glacier bodies lying in a scarcely studied sector of the Central Andes of Argentina. Furthermore, it could be useful for regional and larger-scale assessments. In addition, the simple methodology used is easy to replicate for studying other mountainous regions, as a useful tool to assess periglacial environments in relation to geologic and topographic settings.

## 2 Regional Setting

The study area lies on a northern sector of the Central Andes, along the Argentinean border with Chile, where the headwater sources of the Blanco river basin are located. These hydrological systems feed north-western tributaries of the San Juan river basin, one of the two most important rivers of the province, which have a total mean annual discharge of 56 m³/s in Calingasta, Province of San Juan. The study region belongs to Salina river basin and is delimited by the Blanco River watershed, including the De Las Lagunas or De Las Salinas, Bagres and Frío rivers basin (Fig. 1). The Salinas river basin spreads over an area of 630 km2, with a total perimeter of 155 km. Its basin closing point is located at the geographical coordinates: 31° 21′30′′ and 70° 23′, the point where the Salinas river flows into the Blanco river.

### 2.1 Climatic and geomorphological setting

The climate of this region is typically that of the South America Arid Diagonal. Further, the prevailing climate of this region is semi arid because it lies on the rain-shadow flanks of the Andean Range. Warm moist air rises through Andes orography



which acts as a barrier to moisture transfer from the Pacific (Strecker et al., 2007). Therefore, upon precipitations on the Chilean side, a Föhn wind, with its characteristic dry and warm effects starts blowing on the leeward down-slopes of the Andes mountain ranges. Regionally, this hot and dry wind is called Zonda, and it arises on the lee-side and increases its power on its down slope run, from west-northwest to east-southeast blowing direction. Annual moisture input to the region is highly variable, with close relationship to the state of El Niño Southern Oscillation (or ENSO) occurrence (Masiokas et al., 2006 and Corripio et al., 2007). During the warm ENSO phase, higher temperatures promote the occurrence of snow storms on the high Central Andes ranges, a phenomenon that, altogether with a slight increase in cloudiness, allows to constantly keep low albedo values in this region. These effects decrease the ice-melting process while it increasing the accumulation of glacier mass. This, in turn, renders positive annual ice-mass balance values, or at least less negative ones during ENSO years (Leiva et al, 2007). The precipitation input is also dependent on the wet atmospheric south-east mass from the Atlantic winds (Bolius et al. 2006; Jenk et al., 2015). Three weather stations placed in the study area, specifically two at El Altar and one at Los Azules mining projects, have allowed to define a mean annual air temperature (MAAT) of 2.4°C at 3.375 m.a.s.l. (Schreiber 2015).

The maxim Pleistocene glaciations have modelled the relief. Therefore, in the study area various landforms caused by glacial erosion are found, e.g.: glacial striations, U-shaped and asymmetric valleys, fleecy rocks, erratic blocks, circuses, rams, truncated spurs, edges, tarns, horns, hanging or suspended valleys, among other features. Accumulation landforms conforming glacial deposits, such as moraines, erratic blocks or another tills deposit are frequently found as well. In current times, the glacial landscape is being modified and re-weathered by other process, entailing gravitational, fluvial, aluvial and periglacial processes. As a result, the prevailing current process is clearly noted as one of waste-mass buildup and periglacial phenomena. As regards glaciers, there is a striking number of active, inactive and fossil rock glaciers, as well as other periglacial landforms such as protalus ramparts, solifluction hillsides, stones rings and belts, stratified slope waste deposits, convex-concave debris-mantled slopes and polygonal ground (Fig. 2). The lakes and lacustrine deposits observed are associated to moraines and wasted mass deposits. Below 3.200 m.a.s.l the hydrological system shows a particular occurrence of the high andean wetlands known as vegas, with scrub and hard grasses growing on the foothills and lagoons systems.

## 2.2 Geological setting

The structural, topographic, stratigraphic and geological aspects are highly dependent on the geometry of the Wadatti Benioff zone (Smalley and Isacks, 1990), where the interaction between Southamerican and Nazca tectonics plates take place. This Andean range sector is called the Flat Slab area, because the Nazca plate has a very low subduction angle, smaller than 5° (Isacks y Barazangi, 1977). Currently, over the geological province Cordillera Frontal (Groeber, 1951) this condition is reflected on the surface by an indirect fault system and a folded thrust belt system (Ramos et al., 1996), being this one of the most actively seismic area of Argentina.





In the study area, Eo-Paleozoic rocks deposits are conformed by strongly deformed Intrusive Rocks (DPig). Overlying the Paleozoic rocks, the mesozoic deposit are found, which are mainly represented by the Choiyoi Group (Rolleri y Criado, 1970). This group is conformed by Permian-Triassic volcanic and pyroclastic rocks (PTrv) of alkaline composition, i.e.
basalt, andesite, dacite and rhyolite. The upper section of the group is characterized exclusively by aciditic association of intrusive and vulcanic rocks (PTrg), consisting of gabbros, granodiorites and tonalites (Mpodozis y Ramos 1989). There also is a Jurassic and Lower Cretaceous marine sedimentary sequence (Jk-ism). The cenozoic deposits are represented by Paleogene rocks (EOp and OMp) in close relation with volcanic and plutonic events (Llambías y Malvicini, 1966) There is a wide development of Neogene sedimentary and volcanic sequences (Ms and Mv) of the Lower Miocene. Finally, quaternary
deposits overlie the valleys surface (PlQs), consisting mainly on glacial accumulation, a witness of Pleistocene glacial advance and retreat sequences.

## 3 Material and Methods

In order to analyze the influencing parameters on periglacial environment, the first stage of the research entailed delineating a detailed rock glaciers inventory through a body of geomorphological information. Second a database was performed and in
a third stage entailed designing the inventory database and performing the statistical analysis for rock glacier distribution.

### 3.1 Rock Glaciers Inventory and Characterization

The rock glaciers inventory has been framed within the corresponding hydrologic system for each rock glacier. Drainage network and watersheds mapping was made using a digital elevation model provided by ALOS Palsar AP_07965_FBD_F6550 with 12,5 x 12,5 m spatial resolution. Dataset: ASF DAAC 2015, ALOS
PALSAR_Radiometric_Terrain_Corrected_Hi_res; Includes Material © JAXA/METI 2007. Accessed through ASF DAAC, https://www.asf.alaska.edu 16 May 2016 DOI: 10.5067/JBYK3J6HFSVF. Rock glaciers identification, mapping and inventory was based on Satellite Multi-spectral Images provided by Terra/ASTER; CBERS2B/HRC; Sentinel 2 and SPOT5/HRG2, provided by CONAE (Comisión Nacional de Actividades Espaciales, Argentina) © CNES 2014, Distribution Spot Image S.A. The landforms were manually (on-screen) digitized as vectors, using the projection UTM zone
19 south and WGS84 datum, based on geomorphological and geological information. In general, the methodology of manual digitizing renders better results when used on this kind of landforms (Stokes et al., 2007). On Frio River basin, a detailed field control was made, including periglacial landforms ocurring on Los Patos Norte river, on La Embarrada river and throughout the course of Frío river.



### 3.2 Inventory Database

The designed database included basic descriptive information (basin, satellite and sensor used, type of rock glacier and lithology of the bedrock); localization (latitude and longitude) and physical parameters (Elevation, Slope, Aspect and Potential Incoming Solar Radiation). The database is aimed at compiling the overall information obtained for each periglacial landform, so as to allow studying their relationships and their interpretation.

According to their thermodynamic equilibrium with the environment, rock glaciers were divided into active, inactive and
fossil features. This classification was based on indirect geomorphological criteria identified in satellite images, and complemented with geomorphological and geophysical field information. A key geomorphological indicator that allows discriminating active, inactive and fossil rock glaciers is the evidence of good development and conservation of ridges and furrows creeping on their landform' surface, as a proof of on-going movement. Active rock glaciers show evidence of movement and current geocryological processes occurrence, while inactive and fossil rock glaciers show evidence of past
movement. Inactive rock glaciers still keep the form and ground ice; they do not show a notable development ridges and furrows on the surface. In contrast, the frequent features are the flat surfaces with thermokarst depressions. Other important geomorphological criteria is to ponder the steepness of the front talus slopes. On such a sense, Ikeda and Matsuoka (2006) have considered the limit for active and inactive rock glaciers as of 35° talus slopes, which are angles greater than the angle of repose, and are a sign that the front is advancing. Fossil rock glaciers tend to have gentler slopes and a major degraded
topography. Vegetation presence on the rock glacier surface or talus is an indicator of ice melt and current instability with the environment. Active rock glaciers are characterized by exposure of fine debris at the talus front and large block accumulation at the talus bottom (Wahrhaftig and Cox, 1959; Haeberli, 1985; Roer and Nyenhuis, 2007). Field controls were conducted by late summer and early autumn (March and April), when the active layers show greater development. Field geomorphological studies allowed a better characterization of the periglacial landforms.

The lithology predominant in each rock glacier was calculated in term of areas and account of pixels, using a geologic map based on publications of the Geological and Mining Service of Argentina Republic (SEGEMAR, 2008). The geological formations of the study area were grouped into nine units.

Physical parameters were calculated through the open-source Quantum GIS, GV GIS, SAGA GIS and Kosmo GIS softwares. Calculated values were: minimum, maximum and medium elevation, medium slope, medium aspect and medium potential
incoming solar annual radiation for each kind of rock glacier. The lowest and highest values for each landform are directly calculated with the maximum and minimum raster model pixel values for each landform, while the mean value is computed as the average of all pixels on the raster file. For analyzing the rock glacier's attitudinal distribution, the study area was divided into 5 main categories, called attitudinal belts, on the basis of minimum and maximum landforms elevation values. Medium Slope and Aspect measurements were obtained on SAGA and Gv GIS respectively. Medium aspect results are



circular parameters; therefore, their radian-equivalence was calculated and the sine and cosine values of grid orientation were computed. Finally, the medium aspect (in degrees) was obtained by calculating the arctangent value (Paul et al., 2010).

Potential incoming solar radiation is a holistic parameter which consist in a combination of other raster information such as digital elevation model, sky view factor, latitudinal situation (solar declination and position), slope, aspect, solar constant and heliophany (sunshine hours at each measurement point). The chosen solar constant was 1.367 W/m². This constant is the value assigned by the World Radiation Reference Centre (WRRC) proposed by the WMO (World Meteorological Organization). The calculations for annual potential solar radiation was carried out in SAGA GIS, using ALOS PALSAR DEM. The calculations were performed for a full year, every 7 days in periods of 1.5 hours. In the model, both the atmospheric altitude and the vapour pressure readings were taken into account. The measuring unit for output radiation raster is watt-hours per square meter (W.hr/m²).

### 3.3 Statistical Analysis

In order to discriminate the areas covered by rock glaciers in relation to the different categories of analyzed parameters, the overall regional trends and the local parameters controls, a frequency ratio method was applied. Parameter categorization or classification was based on previous rock glacier distribution analyses. We chose representative values for each category.

The frequency ratio is the probability ratio for an occurrence vs the probability of a non-occurrence for a given attribute (Bonham Carter, 1994). Therefore, the frequency ratio (Fr) can be calculated through the following Eq. (1).

$$F_r = \frac{\frac{N_i}{N}}{\frac{S_i}{S}} \qquad (1)$$

Where S is the total number of pixels; N is the number of pixels with rock glacier occurrences; Si is the number of pixels, being i the factor or variable; and Ni is the number of pixels in which the rock glaciers occurred for the i-attribute or factor. If Fr is greater than 1, it means a higher correlation. A value smaller than 1 means lower correlation.

### 4 Results

#### 4.1. Periglacial Landforms Inventory and Database

The study area covers about 630 km², showing a widespread development of periglacial landforms. Rock glaciers occupied 3, 25 % of said area (Fig. 3).

Nine different geological formations were identified which are spread over different relative areas (Table 1). Geologically, the study area could be divided into six longitudinal north-south belts, separated one another by regional structures. We analysed only the lithology influence over different periglacial landforms (Fig. 4).

The attitudinal distribution varies between 2.940 m.a.s.l. and 4.750 m.a.s.l. Rock glaciers' attitudinal distribution was divided into 4 attitudinal belts, based on minimum and maximum elevation values.

Belt 1- Seasonal Frost Belt: It is the lowest attitudinal belt, conformed by areas below 3.300 m.a.s.l. (lower intact rock glaciers attitude).

Belt 2- Unstable Periglacial Environment Belt: Areas with elevations between 3.300 and 3.690 m.a.s.l (lower elevation of inactive and active rock glaciers)

Belt 3- Current Periglacial Environment Belt: It include areas between the lower and upper active rock glacier elevations. This belt was subdivided into three parts: the lower, middle and upper Current Periglacial Environment Belt, for more detailed attitudinal distribution study (elevation intervals: 3.690-3.868; 3.868-4.047 and 4.047-4.225 m.a.s.l.)

Belt 4- High Periglacial Environment Belt: Include areas with elevation >4225 m.a.s.l., generally mountain peaks, where landforms lie on the highest slopes.

The comparative analysis between attitudinal distribution and spatial arrangement, in terms of longitude and latitude, has not shown significant discrepancies, because it is a local study. Nevertheless, a notable increase towards the east was detected, which is in concordance with the rising altitude of the mountain ranges. That is, in general, on the western side fossil or inactive rock glaciers are predominant, while in highest range areas, there is a larger number of active rock glaciers (Fig. 5).

The mean slope gradient in rock glaciers and protalus ramparts is 19,5 %, whereas intact rock glaciers have mean slopes ranging between 8,5 y 28,5 %. This shows that a very high slope (greater than 30%) is not related to rock glacier development and conservation. Protalus ramparts show a maximum slope of 44,8%. Finally, fossil rock glaciers show a 3,9 % slope (Fig. 6).

The aspect analysis shows that active rock glaciers are governed by clear south and south-west trends. A peak having an eastern orientation appears due to the high Bagres Range which follows the rock glacier development on this direction. North-faces present practically no development of active periglacial landforms. Inactive rock glaciers feature predominant development aspects over the south-face of mountains and, in second order, on the east and west faces, as shown mainly on the Bagres and La Totora ranges. The aspect of fossil rock glaciers has a clear trend toward southeast, while protalus ramparts show a highly diverse characterization. Practically on all direction this type of feature is present, though west facing slopes are the most common ocurrences and north faces are less developed (Fig. 7).



The reading of Potential Direct Annual Radiation shows a minimum of 660 Wh/m$^2$ and a maximum of 2.600 Kwh/m$^2$ for the study area, while the average values for periglacial landforms are 1.800- 1.900 kWh/m$^2$. Values for active rock glaciers lie

between 857 and 2.322 Kwh/m$^2$. The potential incoming solar radiation has provided good estimates because the areal distribution of the periglacial landforms is irregular in the valleys, and this seems to be well correlated with aspect. This is clearly noted on south facing slopes, lower lying terrain and shade-casted areas which allow rock glaciers to develop at lower elevations. However, Potential Incoming Solar Radiation estimates for intact rock glaciers are lower than those for fossil rock glaciers and protalus rampart. Over active rock glaciers, the radiation readings show a bit less solar exposure, and this

fact is in a clear relation with the aspect values. Over south faces slopes there is a minor heliophany which decreases the solar radiation and allows rock glacier to develop or to be preserved furthermore. However, the potential incoming solar radiation values for intact rock glaciers are lower than those for fossil rock glaciers and protalus ramparts. On active rock glaciers, the radiation values shows bit less solar exposure.

### 4.2. Statistical Analysis

The frequency ratio (Fr) analyses over active rock glaciers was calculated for each natural parameters previously analysed. It demonstrates that elevation is the most important parameter, showing the highest frequency ratio value of 6,2 for elevations above 4048 m.a.s.l.. The influence of lithology is the second most important parameter with Fr=4,6 and 2,5 for Oligocene - Lower Miocene Intrusive Rocks (OMp) and Permian – Triassic Volcanic, pyroclastic and sedimentary rocks (PTrv), respectively. OMp consist in porphyry and subvolcanic outcrops, while PTrv are mainly compounded by basalt, andesite,

dacite and rhyolits rocks. Third, active rock glaciers are controlled by aspect, showing the maximum values of frequency ratio (Fr=2,66) on the south faces. Potential incoming solar radiation and slopes show lower influence; however slopes between 11 and 20% and values of radiation between 856.789 to 2.322.270 Wh/m$^2$ show frequency ratio values greater than 1 (Table 2).

### 5 Discussion

From the analyzed data and the computed results, it may be stated at first view that periglacial landforms are significantly affected by lithology. Previous similar studies, such as Johnson et al (2007), propose that lithology could affect rock glacier distribution through influences on hydrology and thermal ventilation, which is a possibility to consider for the studied area. French (2007) considers the geological information an important factor, because lithology controls the capacity of rock glaciers to develop or conserve; while Ikeda and Matsuoka (2006) show that pebbly and boulder material has an influence on

the degree of ridges and furrows developed on periglacial landforms surfaces.

Development and conservation of active rock glaciers, also proved to be strongly influenced by their attitudinal distribution. We divided the area into three zones: the lower, middle and upper current periglacial environment belt. The highest belt



(4.047-4.225 m.a.s.l.) present suitable conditions for active rock glacier development and conservation. However, above this current periglacial environment there are some inactive and also fossil rock glaciers, thus indicating that not all of the area above 3.690 m.a.s.l. features current permafrost conditions; on the contrary the unfavourable conditions for active rock glacier conservation and development prevail in the lowest belt (3.690-3.868 m.a.s.l.). Different authors have indicated that the lower attitudinal limit of active rock glaciers is representing the lower limit of discontinuous mountain permafrost (Brenning, 2005). This is consistent with the lower limit of 3.725 m.a.s.l estimated from geophysical surveys in three rock glaciers over Frío river basin (Forte et al., 2013; Villarroel et al., 2016). Between the lowest active and inactive rock glaciers elevations (3.690 to 3.300 m.a.s.l.) the current periglacial environment is subject to degradation processes, according to Schreiber (2015) the mean annual air temperature (MAAT) is 2.4°C at 3.375 m.a.s.l. in the region. However, the protalus rampart existence on this belt demonstrated that current periglacial local processes are ongoing locally even this unstable periglacial environment belt.

## 6 Conclusions

We have demonstrated in this work the important role of lithology in rock glacier development which also implies that altitude is not the only variable controlling development and conservation of periglacial environment landforms. Sedimentary, pyroclastic, subvolcanic and volcanic or plutonic with hydrothermal alteration rocks seem to be more appropriate material for rock glacier growth and development. However, we have noticed that inactive rock glaciers are more common over these terrains. Therefore, these lithologies at the same time, are prone to higher degradation. This is consistent with the easiness of the matter for weathering off, along with their natural porosity which allows greater water infiltration and weaknees which furthermore, allows for greater weathering and erosion processes, mainly cryoclastism. Nevertheless, intrusive and granitic rocks seem to be a better lithology feature for active rock glaciers conservation. In the studied lithology there is a significant number of active rock glaciers, though fewer inactive rock glaciers, even at similar or worst conditions of slope, aspect, elevation and radiation. Therefore, the study points out some intriguing behaviour as a lithology, because it has a good capacity for rock glaciers development, but is inadequate for rock glaciers conservation under climate changes.

Intact rock glaciers and protalus ramparts are indicators of periglacial environment. However, we had called 'current periglacial environment belt' to the area above the lower active rock glaciers elevation (3.690 m.a.s.l.). Between the lowest active and inactive elevations (between 3.300 to 3.690 m.a.s.l.), the current periglacial environment is subject to degradation processes. Below it, the current periglacial processes have disappeared. However there is a large number of fossil rock glaciers and other periglacial landforms that are indicating a favourable paleo environment for cryogenic process occurrence. Active and inactive rock glaciers have a mean attitudinal of 3.960 and 3.950 m.a.s.l. respectively, and are mainly found above the lowest active rock glacier elevation (3.690 m.a.s.l.). About 90 % of inactive and 61 % of fossil rock glaciers are

located completely above this elevation. This fact demonstrates that altitude is not the only variable controlling the development and conservation of periglacial environment landforms.

Other important influence parameter is the orientation of the slopes (aspect), it is mainly expressed by the irregular distribution over the valleys of periglacial landforms, with a remarkable major development and conservation on south-facing slopes. The study area is located about 1.000 km at south of the Capricorn Tropic where the sun is perceived to be directly overhead (on the zenith) of the southern summer solstice. This subtropical situation is one of the main influencing factors for the rock glaciers development.

The semi-arid climate, characterized by receiving precipitation rates below those of potential evapo-transpiration, follows a high potential solar radiation on the entire studied area. However, intact rock glaciers seem to be developed over an ample range of solar radiation conditions. This is because the topography is not so sloped and heliophany is similarly rated over the entire studied area. Only aspect and sun inclination are the main factors controlling the potential incoming solar radiation.

The larger number of rock glaciers is found in the Frío river basin (n:15). Whereas the basin with the major development of
active rock glaciers in relation to the basin area is found on the Bagre river watersheds (n:12), over this basin the most favourable lithology and the highest peaks are found. The ratio between area occupied and amount of rock glaciers is similar to that for intact rock glaciers. However, fossil rock glaciers show a tendency to use larger areas, in contrast with protalus ramparts, whose number is much greater than the occupied area. In several regions, was noted a continuos disposition between newer active rock glaciers that overlap above the inactive ones. A similar situation has been observed with inactive
above fossil rock glaciers.

The results presented in this work constitute an important progress in documenting number, features and distribution of rock glacier bodies lying in this important sector of the Central Andes. Furthermore, the methodology presented here is easy to replicate and may be applied to other mountainous regions to assess periglacial environments in relation to geologic and topographic settings.

**7 Acknowledgements**

This research was supported by CONICET (Consejo Nacional de Investigaciones Científicas y Técnicas) research scholarship, Argentina. SPOT multi-spectral satellite images were obtained thanks to an agreement between UNSJ (Universidad Nacional de San Juan) and CONAE (Comisión Nacional de Actividades Espaciales, Argentina). Ana Forte is grateful to Roberto Medrano, Flavia Tejada, Alejando Lopez, Pablo Gutierrez and Florencia Gerarduzzi for field work
collaboration.



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





**Figure 1: The study area represents about 13% (630 Km2) of the Blanco River Basin (4.861 km2).**



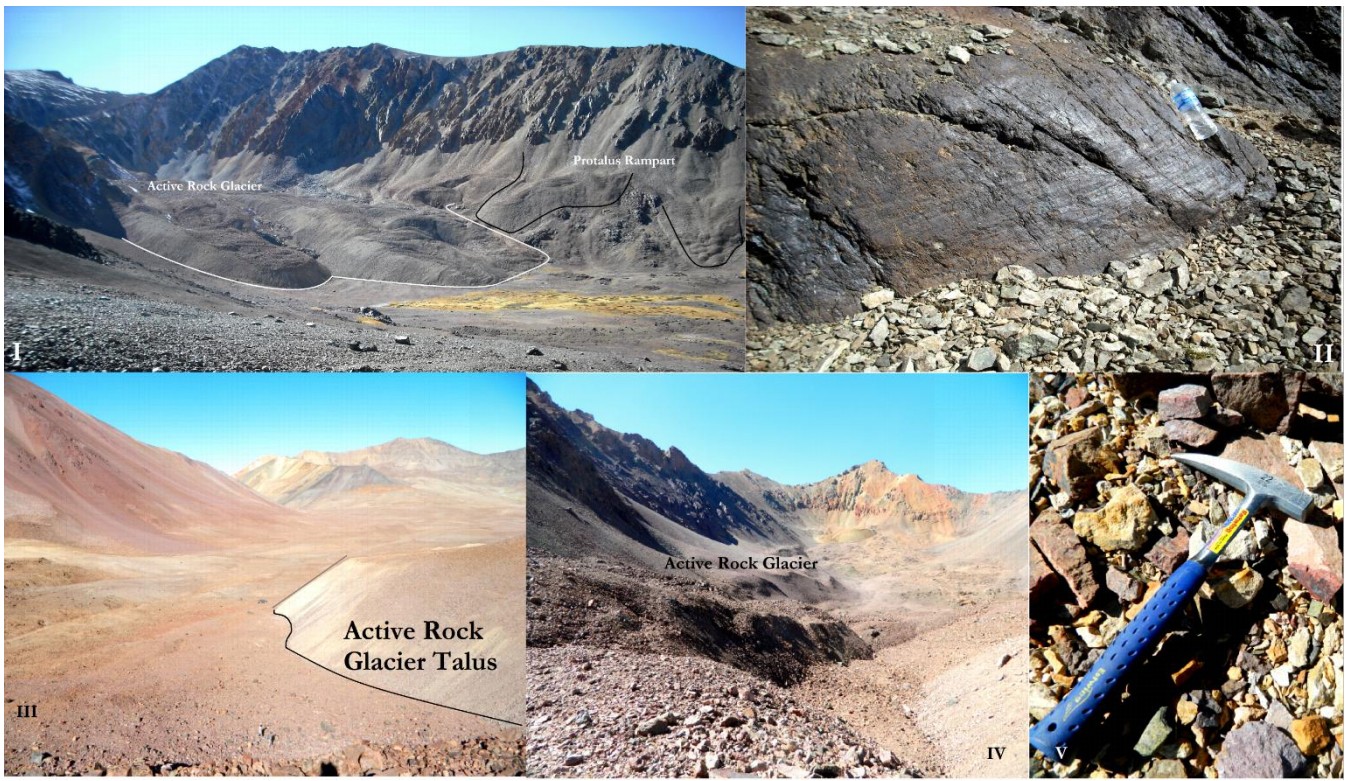

**Figure 2: Current geomorphological processes superimposed to glacial accumulation and erosion landforms . I- Active Rock Glaciers and Protalus Ramparts over an old circuses and U-shaped glacial valleys. II – Glacial striations erosion and cryoclastism debris. III- Active Rock Glacier Talus. IV- Active Rock Glacier. V- Cryoclastism.**





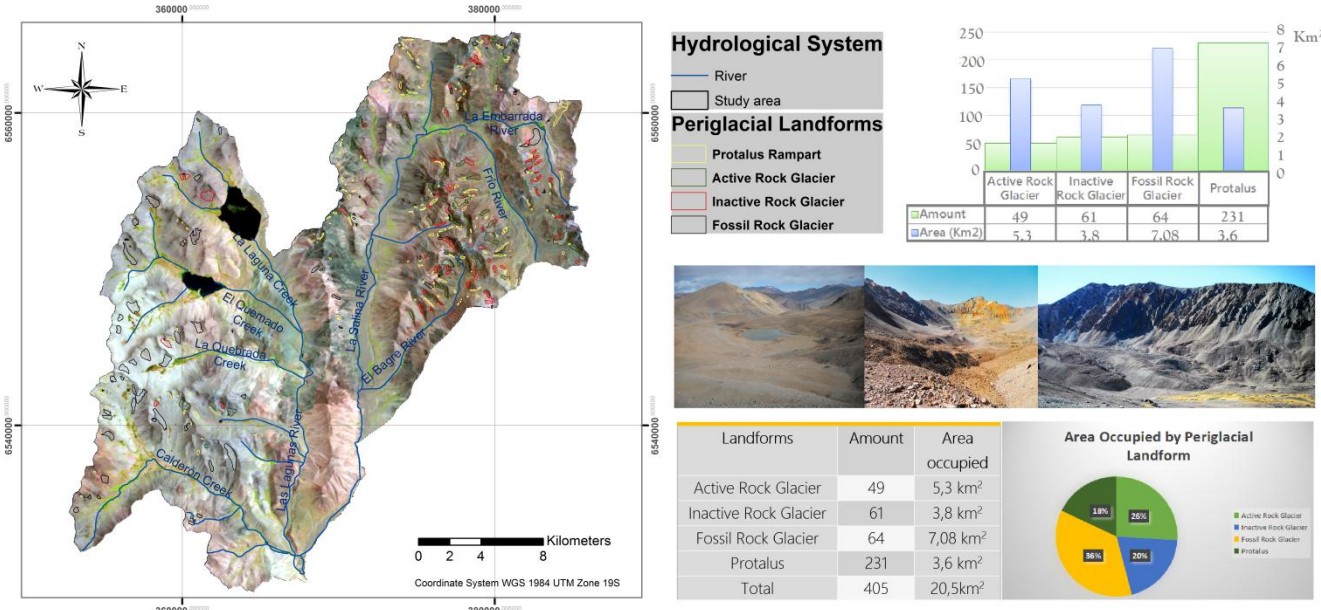


**Figure 3: Rock glaciers inventory details.**

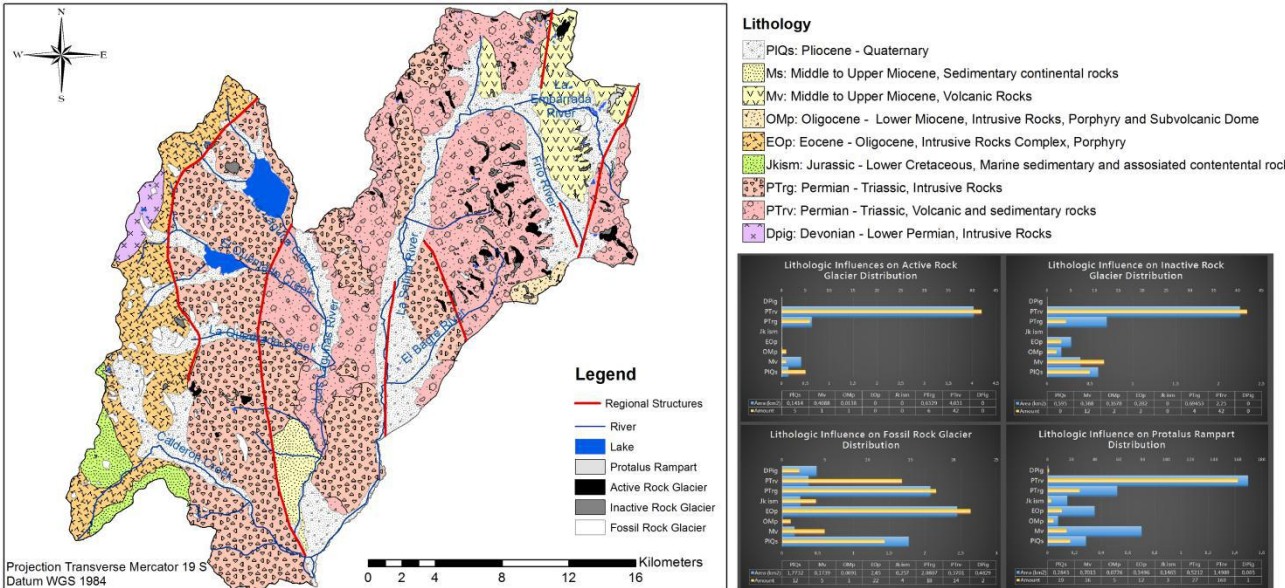

**Figure 4: Geological map based on SEGEMAR GIS (2008) and rock glaciers inventory. Lithologic influence for different periglacial landforms.**




**Figure 5: Attitudinal and spatial distribution of periglacial landforms and hypsometric analysis.**



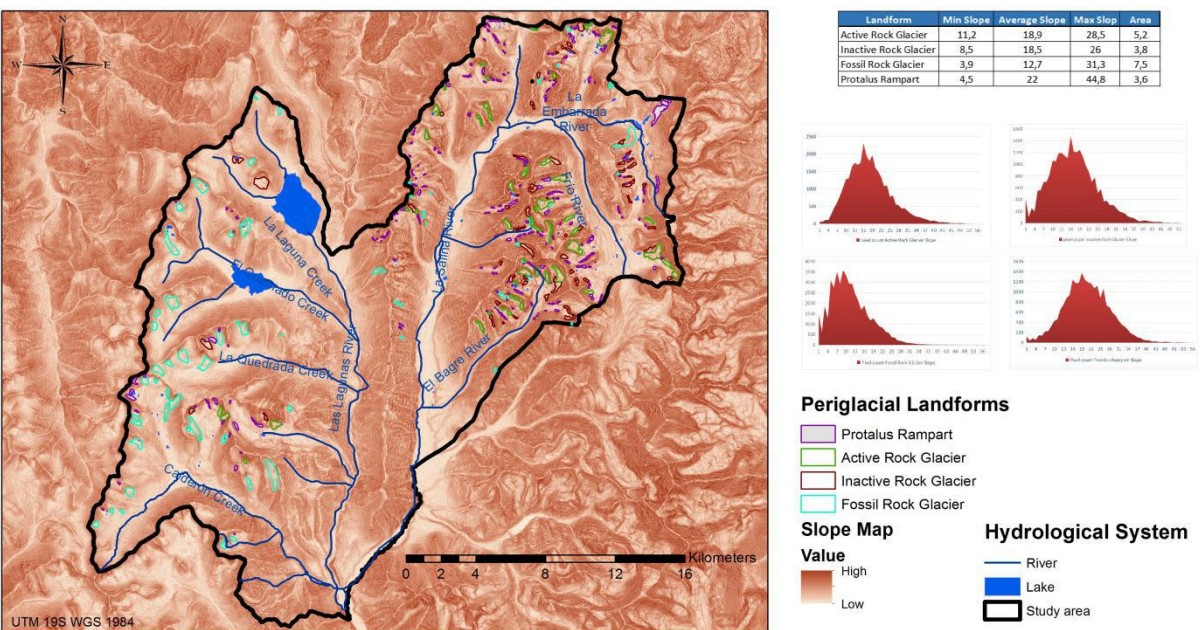

**Figure 6: Slope map. Table of mean slope (upper right corner) and charts with slope values over rock glaciers and protalus ramparts.**

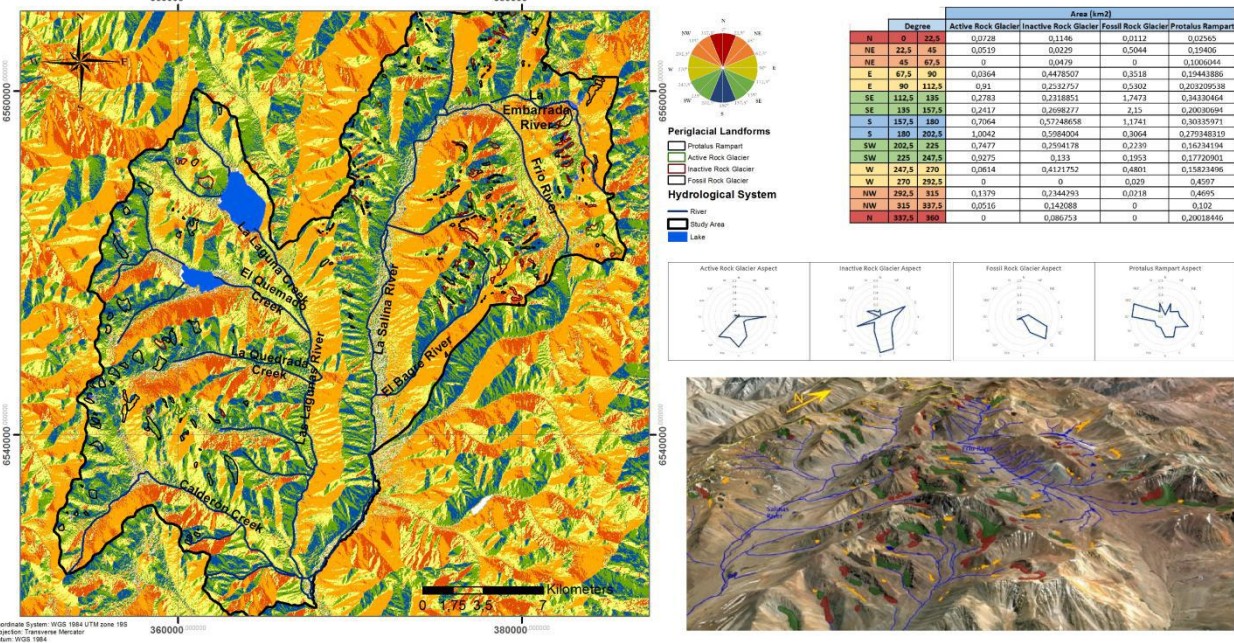


**Figure 7: I- Aspect map (in degrees). Red and orange colours represent the warm north face slopes; yellow represents the west and east faces; while blue and light blue show the cold south faces. II- Table of Categories for Aspect Classification. III- Aspect chart for each landform. IV- 3D View (from Google Earth) of studied landforms and surface terrain. The green features represent the Active Rock Glaciers, which are located mainly on over south faces.**




**Figure 8: Potential Incoming Solar Radiation (Wh/m².).**

**Table 1: Geological Units.**

| Symbol | Age | Description | Area (km²) |
|--------|-----|-------------|-----------|
| PlQs | Pliocene - Quaternary | Sedimentary deposits | 2.7939 |
| Mv | Middle to Upper Miocene | Volcanic Rocks | 1.672 |
| Ms | Middle to Upper Miocene | Sedimentary clastic rocks | 0.614 |
| OMp | Oligocene - Lower Miocene | Intrusive Rocks, Porphyry and subvolcanic dome | 0.2683 |
| EOp | Eocene - Oligocene | Intrusive Rocks Complex, Porphyry | 3.0816 |
| Jk ism | Jurassic - Lower Cretaceous | Sedimentary clastic rocks | 0.4035 |
| PTrg | Permian - Triassic | Intrusive Granitic acid Rocks | 3.92933 |
| PTrv | Permian - Triassic | Volcanic, pyroclastic and sedimentary alkaline rocks | 8.15 |
| DPig | Devonian - Lower Permian | Intrusive basic Rocks | 0.4879 |

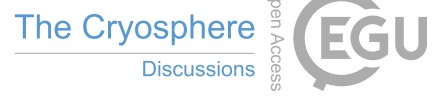

**Table 2: Ratio Frequency Analyses.**

| Factor | Class | Nº de pixels showing active rock glaciers occurrence[a] | % of pixels showing rock glaciers occurrence[b] | Pixel in domain[c] | Pixel %[d] | Frequency ratio[e] |
|---|---|---|---|---|---|---|
| Elevation [m.a.s.l.] | 2950-3299 | 0 | 0,000 | 723601 | 18,104 | 0,000 |
| | 3300-3689 | 0 | 0,000 | 1252217 | 31,330 | 0,000 |
| | 3690-3867 | 9057 | 27,188 | 1178607 | 29,488 | 0,922 |
| | 3868-4047 | 12186 | 36,580 | 427470 | 10,695 | 3,420 |
| | 4048-4225 | 12070 | 36,232 | 232191 | 5,809 | 6,237 |
| | 4226-4763 | 0 | 0,000 | 182801 | 4,574 | 0,000 |
| Aspect [degree] | 338-22 | 703 | 2,110 | 504867 | 12,632 | 0,167 |
| | 23-67 | 1250 | 3,752 | 497005 | 12,435 | 0,302 |
| | 68-112 | 3819 | 11,464 | 603825 | 15,107 | 0,759 |
| | 113-157 | 6286 | 18,870 | 544826 | 13,631 | 1,384 |
| | 158-202 | 9124 | 27,389 | 411352 | 10,292 | 2,661 |
| | 203-247 | 8039 | 24,132 | 494739 | 12,378 | 1,950 |
| | 248-293 | 2953 | 8,864 | 545267 | 13,642 | 0,650 |
| | 294-337 | 1139 | 3,419 | 395006 | 9,883 | 0,346 |
| Slope [percentage] | 0 - 10 | 6109 | 18,338 | 1072973 | 26,845 | 0,683 |
| | 11 - 19 | 17930 | 53,823 | 1219425 | 30,509 | 1,764 |
| | 20 - 35 | 8525 | 25,591 | 1495870 | 37,426 | 0,684 |
| | 35 - 75 | 749 | 2,248 | 208619 | 5,220 | 0,431 |
| Lithology | Pliocene-Quaternary - Sedimentary deposits | 900 | 2,702 | 855951 | 21,415 | 0,126 |
| | Middle to Upper Miocene - Volcanic Rocks | 2570 | 7,715 | 192497 | 4,816 | 1,602 |
| | Middle to Upper Miocene - Sedimentary rocks | 0 | 0,000 | 73069 | 1,828 | 0,000 |
| | Oligocene to Lower Miocene - Intrusive Rocks | 884 | 2,654 | 22802 | 0,570 | 4,651 |
| | Eocene to Oligocene - Intrusive Rocks | 0 | 0,000 | 374766 | 9,376 | 0,000 |





| | | | | | | |
|---|---|---|---|---|---|---|
| | Jurassic to Lower Cretaceous - Sedimentary rocks | 0 | 0,000 | 90136 | 2,255 | 0,000 |
| | Permian to Triassic - Intrusive rocks | 4050 | 12,157 | 1227578 | 30,713 | 0,396 |
| | Permian to Triassic - Vulcano-sedimentary rocks | 24909 | 74,773 | 1117440 | 27,958 | 2,674 |
| | Devonian to lower permian - Intrusive rocks | 0 | 0,000 | 42648 | 1,067 | 0,000 |
| Potential Solar Radiation [WH/m2] | 660 - 856788 | 0 | 0,000 | 743104 | 18,592 | 0,000 |
| | 856789 - 1839402 | 18568 | 55,738 | 1236587 | 30,939 | 1,802 |
| | 1839403 - 2322270 | 14745 | 44,262 | 991562 | 24,808 | 1,784 |
| | 2322271 - 2598972 | 0 | 0,000 | 1025634 | 25,661 | 0,000 |

a = Total number of pixels showing active rock glaciers occurrence = 33.313

b = (a/33313)*100

c = Total number of pixel in domain = 3.996.887

d= (c/3996887)*100

e= b/d