# Peer review of "Impact of natural parameters on rock glacier development and conservation in subtropical mountain ranges. Northern sector of the Argentine Central Andes"

_The Cryosphere, 2016_

## Referee Comment (RC1) · Anonymous Referee #1 · 20 Mar 2017

Dear Editor,

I have reviewed the manuscript "Impact of natural parameters on rock glacier development and conservation in subtropical mountain ranges. Northern sector of the Argentine Central Andes" by Forte et al. with interest. The authors provide a new inventory of rock glaciers and protalus rampart for subtropical mountain ranges in the northern sector of the Argentine Central Andes. The authors mainly describe the characteristics of the inventory and finally apply a statistical approach (frequency ratio) to investigate controlling factors. The topic is of interest for The Cryosphere and the statistical analysis demonstrate that elevation, lithology and aspect are also dominant controls for subtropical mountain ranges. Nevertheless, my review pointed out a number of serious scientific issues, which are listed in the following general comments and in the following detailed comments section. In my point of view, these points must be carefully addressed before the manuscript can be considered for publishing in a high-level journal as The Cryosphere. Thus, I suggest reconsideration of the manuscript after major revisions.

GENERAL COMMENTS

1) The authors provide a new inventory of rock glaciers and protalus rampart and use it for further analysis. The inventory bases on a singular survey and therefore can be used to describe the occurrence of rock glaciers and protalus rampart, but not to describe the development and conservation which is the intention of the authors (see title, abstract and elsewhere).

2) The authors formulated the overall aim of the study, but a clear research question in the introduction is missing. An overall clear structure would facilitate reading the manuscript.

3) The rock glaciers and protalus rampart inventory for this region is novel. The resulting inventory database includes basic descriptive information, localization, physical parameters and a classification (dividing rock glaciers into active, inactive and fossil). The generation of the inventory and inventory database need to be explained in more detail. A validation and uncertainty assessment are missing and would be very important, because the statistical approach bases on it. Especially the detection and classification leaves a lot of room for interpretation. Further it is not clear how the observed features are assigned to protalus rampart or rock glaciers (what are the criteria for distinction?).

4) The description of the statistical analysis should be extended and formulated more clearly. Please ensure that the mathematical variables are unique and correctly defined.

5) The discussion section is much too short and should be extended adequately. There are many open questions: How do rock glaciers in subtropical mountain ranges differ from other regions? How reliable/robust are the results of the statistical analysis? And many more...

6) There is a lot of potential to improve the figures with regards to content, graphics and readability. Try to make the content of the figures and the relation between the subfigures clear. Try to avoid repetition of figures (e.g. Fig. 2 and Fig. 3: the relevant pictures are shown twice). I suggest to reduce the content in figures. Putting the maps providing physical parameters and geology in a figure with several subplots facilitates comparison. Try to use appropriate chart types and to avoid duplicated information visualization (e.g. bar chart and table) unless there is real advantage. A consistent style of the figures with less color, better color selection (e.g. colorbrewer2.org) and clear color definition would improve the quality. Be aware of the quality of the figures and the readability of text. Parts in Fig. 1, 3, 4, 6 and 7 are non-readable and need to be guessed.

7) Language and consistency: I'm not a native speaker myself, but spell check and language check would be very helpful. Further be aware of inconsistencies in word/term choice, citing (e.g., p.2, l.4: who is Schrott H.?) and definitions (e.g., mostly rock glacier and protalus rampart are distinguished, but sometimes combined or mixed up).

DETAILED COMMENTS

p. 1, l. 1: The title does not clearly reflect the contents of the study (see General Comment 1).

p. 1, l. 11: Rock glaciers can be a part of periglacial environment, but the occurrence is not necessary and there are many other features.

p. 1, l. 25: Using an inventory based on a singular survey, how do you know that the

rock glacier conservation is mainly controlled by lithology?

p. 2, l. 39: If you mentioned the study area, you would already emphasize that there is no existing inventory/study for your study site.

p. 2, l. 46: How do you know the state of the periglacial environment? A time series of observation/inventories would allow to investigate the evolution.

p. 2, l. 46: Glaciers have a reaction time to climate change of up to several decades.

p. 2, l. 48: I would rather use degradation instead of deterioration.

p. 2, l. 54ff: Is this relevant for this study?

p. 3, l. 67: SW = south west?

p. 3, l. 68f: Change order, first inventory, then statistical analysis...

p. 3, l. 77: This needs to go to the discussion.

p. 3, l. 80: Study site?

p. 3, l. 84: Is there a reference?

p. 3, l. 89: Rephrase this sentence.

p. 4, l. 98: Why is the accumulation of glacier mass relevant? Or do you mean rock glacier?

p. 4, l. 104: I would use rather "shaped" than "modeled".

p. 4, l. 115: As you are only considering lithology and not geology, shorten this paragraph

p. 5, l. 122: confirmed?

p. 5, l.137ff: This paragraph is hard to read and it is not clear what the date of physical inventory is.

p. 6, l. 149: It is not described how the protalus ramparts are distinguished from the rock glaciers.

p. 6, l. 174: What is a "medium" elevation? Do you mean "median" or "mean"?

p. 6, l. 177: What is a "attitudinal distribution"? Do you mean "altitudinal distribution"?

p. 6, l. 178: Why do you have 5 classification?

p. 7, l. 180: For me it is not clear how you calculate a mean aspect. Please explain in more detail.

p. 7, l. 187: Could you please explain in more detail how you calculate the potential incoming radiation?

p. 7, l. 188: How do you know the "vapour pressure"?

p. 7, l. 190: Please explain more precisely.

p. 7, l. 196: For your analysis, you should consider the different classes (e.g. i) of all parameters (e.g. j). Try to use appropriate naming (e.g. classes, parameters, . . .) here and elsewhere in the manuscript.

p. 7., l. 202: Language.

p. 8, l. 207ff: Why 4 belts? The characterization (seasonal frost belt, unstable periglacial environment, . . .) is not only caused by the altitude. And how do you get the boundary values of the belts?

p. 8, l. 222: What is a "slope gradient"? Why do you combine rock glaciers and protalus ramparts here and distinguish them elsewhere?

p. 9, l. 244: Could you show the rock glaciers which are in the altitudinal range 3868-4225 m a.s.l., aspect between 113° and 247°, slope between 11 and 19° as well as "Oligocene to Lower Miocene – Intrusive Rocks" lithology? Are these parameters in combination most appropriate?

[Figure]

p. 9, l. 245: Do you now include protalus ramparts or not?

p. 9, l. 255: This study shows the affect of the different parameters only for rock glaciers and protalus rampart and not for periglacial landforms in general.

p. 9, l. 261: Again, how are you able to provide information on development and conversation of active rock glaciers using an inventory of a singular survey?

p. 9, l. 262: What is with the first belt of before?

p. 10, l. 264: Are inactive rock glaciers really an indication that no permafrost occurs?

p. 10, l. 276: Variable, parameter or factor? Try to be precise and consistent with terms.

p. 10, l. 291: This is a new result and should not be presented the first time in the conclusion. Further, I doubt that the accuracy is such high that you can distinguish 10 m of difference.

p. 11, l. 295: Where/how do you determine the aspect?

p. 11, l. 297: Again new result. . .

p. 11, l. 304: What does "(n:15)" mean?

p. 11, l. 313: Do you consider geology in your analysis?

p. 12, l. 321: Again try to be consistent, even in the References. Once "Arenson, L." and once Arenson, L. U.".

p. 17, Fig 1: Different coordinate systems are used in the different subfigures. There is an error in the coordinates (subfigure left). What is the meaning of the colors? How do the subfigure left and the subfigure top right fit together? Legend of subfigure top right is non-readable.

p. 18, Fig. 2: Where were the pictures taken? What is the relevance of the picture top right?

p. 19, Fig. 3: Try to use better colors to indicate the periglacial landforms. The pictures in the middle right are the same as in Fig. 2. Try to use appropriate chart types and to avoid duplicated information visualization (e.g. bar chart or table, but not both). Legend in subfigure bottom right is non-readable.

p. 19., Fig. 4: Protalus ramparts and rock glaciers are almost not visible and the different landforms are almost not distinguishable from each other. Subfigure bottom right is non-readable.

p. 20, Fig. 5: It is very difficult to take advantage of the colors in the legend ob subfigure top left.

p. 21, Fig. 6: What are the high and low values of the slope map (subfigure top left)? Subfigure top right is non-readable.

p. 21, Fig. 7: I suggest to emphasize the spider chart.

p. 23, Tab. 2: I would suggest to use histogram to visualize the data of this table. How are the classes defined? Sometimes, the classes are not the same as in the Figures/text.

p. 24, l. 470: I guess this should show an example how the values in the column are calculated. Please try to use appropriate variable names (same as in Eq. 7) and give an explanation.

---

## Referee Comment (RC2) · Anonymous Referee #2 · 20 Apr 2017

General comments

The authors present an inventory and spatial anaysis of rock glaciers in a small watershed in the semi-arid Argentinean Andes. The analysis is based on the frequency ratio method. Due to the limited spatial extent of the study area, the methods used and the depth of the discussion, the implications of this study are very limited and could be of interest to researchers with a focus on the semi-arid Andes. The study furthermore suffers from various conceptual and terminological limitations.

In its present form and with this geographical scope, from my view this study seems to

be too limited for publication in The Cryosphere.

Specific comments

L11 "affect", Title and P2L39 "impact", and in other instances the wording indicates a causal interpretation of empirical patterns that is not supported by an in-depth discussion of underlying processes and possible confounders. Descriptive wording (e.g., association, correlation, relationship) should be used where appropriate

L34-36 Proper reference should be made to recent studies focusing on relationships between cryosphere (and mountain permafrost in particular) and hydrology as well as "human activities" (mining) in semi-arid mountain regions.

L46 "unlike glaciers" - Glaciers also show delayed responses (and complex feedback mechanisms) e.g. due to their size or an increasing debris cover thickness.

L48 "faithfully reflect..." - This statement is in clear contradiction to the previous comment concerning the delayed response of rock glaciers to climate change.

L49-50 Wahrhaftig and Cox (1959) reference - references to more recent publications would seem appropriate here

L50-51 "closely related to their equilibrium state" - provide reference

L56 That depends on the type of inactivity; see e.g. Barsch (1996)

L58 protalus rampart - If they can be considered small rock glaciers, why use the concept of protalus rampart? Can protalus ramparts be active, inactive and relict?

L89 To my knowledge, this area is located south of the Arid Diagonal. Please refer to the climatological literature.

L90 Aridity in this region is not primarily due to the "rain shadow" of the Andes. Please refer to the climatological literature.

L93 Is the Zonda reelevant for this study, which focuses on high-altitude regions in the
central part of the Andes, not the valleys and foreland toward the East?

L95 "close" relationship with ENSO is overstated

L97-100 This summary of the relationship between ENSO and glacier mass balance is weak and of little relevance for the present study

L101-103 MAAT for which years? are these years representative?

L104-106 This section overstates the degree to which Pleistocene glaciations have modified the landscape.

L110 "As regards glaciers" - none of the features mentioned in this sentence is considered a glacier

L116-121 Unnecessary structural geological detail

L154ff - Provide references supporting the relationship between form and activity status. Roer and Nyenhuis (2007) provide a comprehensive summary of criteria, which should be considered more carefully. Azócar et al. (2017) in The Cryosphere apply similar criteria to an area in the semi-arid Andes.

L154 "thermodynamic equilibrium" - is it appriopriate to apply this concept here, in particular to inactive rock glaciers that may be degrading?

L192 provide reference(s); is this a state-of-the-art technique? The frequency-ratio method requires splitting of predictors into discrete classes. These classes are poorly justified in this study. Methods such as logistic regression or generalized additive models allow for continuous relationships and are capable of accounting for the influence of each predictor while accounting for the other predictors.

L199 The goodness-of-fit of the model should be assessed using criteria that are to be indicated in this section.

L204-206 This paragraph does not report results

L209-210 How do the authors know that seasonal frost is of particular importance in this elevation range? Rock glaciers and their possible preservation or degradation are not primarily driven by seasonal temperature fluctuations.

L222ff The authors need to distinguish between the distribution of the environmental variables conditional on rock glacier presence (or activity status), and the occurrence of rock glaciers (or their activity status) conditional on these variables. The former way of conditioning does not account for the different frequencies (in the landscape) of, e.g., west-facing and south-facing slopes. Again, a multiple-variable logistic regression model would account for possible confounders.

L247-248 The mentioned Oligocene class represents only 0.57% of the study area; its frequency ratio has insufficient empirical support.

L255-256 The empirical relationship between lithology (and effectively only one of its classes) and rock glacier occurrence can be confounded with other possible factors such as slope angle or solar radiation since the frequency ratio method is unable to separate these empirical relationships. Furthermore, a possible causal link related to specific characteristics of these rocks should be established before claiming that periglacial landforms (or specifically, rock glaciers) are "affected" by lithology.

L276 Much of the material covered in the Conclusions section should be part of the Discussion section where it should be discussed in the context of the available literature; part of this section also simply repeats some of the results and interpretations made before. The current Discussion (considering both sections 5 and 6) of results, their interpretation, uncertainties and limitations of methodology, is currently too weak.

Technical corrections

Terminology and language: - natural parameters -> environmental variables - conservation -> preservation

Additional language editing is required.

---

## Author Comment (AC1) · 19 May 2017

**Reply to Referee #1**

I have reviewed the manuscript "Impact of natural parameters on rock glacier development and conservation in subtropical mountain ranges. Northern sector of the Argentine Central Andes" by Forte et al. with interest. The authors provide a new inventory of rock glaciers and protalus rampart for subtropical mountain ranges in the northern sector of the Argentine Central Andes. The authors mainly describe the characteristics of the inventory and finally apply a statistical approach (frequency ratio) to investigate controlling factors. The topic is of interest for The Cryosphere and the statistical analysis demonstrate that elevation, lithology and aspect are also dominant controls for subtropical mountain ranges.

We would like to thank Referee #1 for emphasizing the importance of the rock glacier and protalus rampart characterization and distribution surveys in subtropical mountainous areas.

Nevertheless, my review pointed out a number of serious scientific issues, which are listed in the following general comments and in the following detailed comments section. In my point of view, these points must be carefully addressed before the manuscript can be considered for publishing in a high-level journal as The Cryosphere. Thus, I suggest reconsideration of the manuscript after major revisions.

The general and detailed comments had been carefully addressed and we would like to thank referee 1 for the contributions.

**GENERAL COMMENTS**

1- The authors provide a new inventory of rock glaciers and protalus rampart and use it for further analysis. The inventor y bases on a singular survey and therefore can be used to describe the occurrence of rock glaciers and protalus rampart, but not to describe the development and conservation which is the intention of the authors (see title, abstract and elsewhere).

In the first general point, Refree1 presents our work, details the general points and makes a first significant observation. Then emphasizes the use of the terms "development" and "conservation", and suggests that we should use the term "occurrence" of rock glaciers and protalus rampart, since our descriptions are limited to that concept. The authors consider that the existence of rock glaciers and protalus rampart are indicators of processes for the development and / or conservation of these landforms. However, we have discussed the proposal again and we will consider limit the use of these terms in the body of the manuscript (title, abstract and others) as Referee #1 has suggested and to raise in the conclusion and discussion the use of these terms more clearly in order to avoid confusion.

We have considered to name the paper: "The influence of environmental variables on rock glaciers occurrence. Argentine Central Andes north sector"

2- The authors formulated the overall aim of the study, but a clear research question in the introduction is missing. An overall clear structure would facilitate reading the manuscript.

In the second point out that although we have formulated a clear research question, it is not clear in the introduction. In this regard we have decided to improve it, in order to facilitate the reading of the manuscript.

The research question is How do some environmental variables influence the occurrence of rock glaciers?

3- The rock glaciers and protalus rampart inventory for this region is novel. The resulting inventory database includes basic descriptive information, localization, physical parameters and a classification (dividing rock glaciers into active, inactive and fossil). The generation of the inventory and inventory database need to be explained in more detail. A validation and uncertainty assessment are missing and would be very important, because the statistical approach bases on it. Especially the detection and classification leaves a lot of room for interpretation. Further it is not clear how the observed features are assigned to protalus rampart or rock glaciers (what are the criteria for distinction?)

We appreciate the consideration of the new scientific information for this region. As for the inventory and database methodology, we recognize that in the manuscript we do not detail enough the way in which manual mapping and digitization was performed. This was due to the existence of a large number of scientific articles based on digital mapping with optical satellite imagery.

- For the rock glacier and protalus rampart Identifying and mapping, was mainly considered geomorphological criteria's, making observations with satellite images and a detailed field control.

- Rock glaciers are distinctive geomorphological landforms, easily identifiable through satellite images with a high spatial resolution (Wahrhaftig and Cox, 1959; Haeberli, 1985; Roer and Nyenhuis, 2007). These may extend from the rock wall, outward and downslope with a moderated slope and a frontal talus with a pronunciated slope. Above the rock glacier surface could be develop furrows and ridges or collapsed structures associated to ice melting can be developed. Intact rock glaciers were differed as active and inactive, using geomorphological criteria as the degree of furrows and ridges development, among others. Active rock glaciers present a better development, also these are characterized by exposure of fine debris at the talus front and large block accumulation at the talus bottom. Other geomorphological criteria used was the angle of the frontal talus slope, which is major on active rock glacier. The existence of vegetation, collapsed structures and flat surfaces are characteristics of inactive rock glaciers. In the Figures 1 and 2 there are two examples of active and inactive rock glaciers mapping.

Fig1: a- Active rock glacier mapping. Geographic coordinates: 31° 9' 45" S and 70° 12' 50" W

b- Field Control. Furrows and ridges over rock glacier surface. Two people as scale.

---

## Author Comment (AC2) · 19 May 2017

**Reply to Referee #2**

**General comments**

The authors present an inventory and spatial analysis of rock glaciers in a small watershed in the semi-arid Argentinean Andes. The analysis is based on the frequency ratio method. Due to the limited spatial extent of the study area, the methods used and the depth of the discussion, the implications of this study are very limited and could be of interest to researchers with a focus on the semi-arid Andes. The study furthermore suffers from various conceptual and terminological limitations.

In its present form and with this geographical scope, from my view this study seems to be too limited for publication in The Cryosphere

Response: The reviewer comments were very useful and enabled us to greatly improve the quality of our manuscript. We have included a multivariate statistical analysis of logistic regression, attached in this document *(see Appendix 1).

The selected study area has no previous studies and we consider that our contribution exceeds regional interests.

**Specific comments**

L11 "affect", Title and P2L39 "impact", and in other instances the wording indicates a causal interpretation of empirical patterns that is not supported by an in-depth discussion of underlying processes and possible confounders. Descriptive wording (e.g., association, correlation, relationship) should be used where appropriate

Response: Thank you for your suggestions. We have made modifications in the wording. Our intention was not to establish a causal interpretation between parameters and rock glaciers occurrence. We hope now is better.

L34-36 Proper reference should be made to recent studies focusing on relationships between cryosphere (and mountain permafrost in particular) and hydrology as well as "human activities" (mining) in semi-arid mountain regions.

Response: Thank you for your suggestions. We have included the following additional references (Azócar and Brenning 2009; Arenson and Jakob (2010) and Brenning and Azócar, 2010).

Azócar, G. F. and Brenning, A., 2009. Hydrological and geomorphological significance of rock glaciers in the dry Andes, Chile (27°–33°S). DOI: 10.1002/ppp.669

Brenning, A. and Azócar, G. F., 2010. Minería y glaciares rocosos: impactos ambientales, antecedentes políticos y legales, y perspectivas futuras. Revista de Geografía Norte Grande, 47: 143-158.

Arenson, L. and Jakob, M., 2010. The significance of rock glaciers in the dry Andes - A discussion of Azócar and Brenning (2010) and Brenning and Azócar (2010). Permafrost and Periglacial Processes, 21 (3): 282-285.

L46 "unlike glaciers" - Glaciers also show delayed responses (and complex feedback mechanisms) e.g. due to their size or an increasing debris cover thickness.

Response: Thank you for your suggestions. We meant here to emphasise that rock glaciers respond with a major delay than glaciers, even debris covered glacier.

L48 "faithfully reflect..." - This statement is in clear contradiction to the previous comment concerning the delayed response of rock glaciers to climate change.

Response: Thank you for your suggestions. This is correct, the response has a delay. But we rather assume here that rock glaciers have high sensitivity to environmental changes. We modified the sentence to improve it

*Unlike glaciers, rock glaciers show a greatest delay in response to climate change and are conditioned by different parameters which control the velocity and degree of development, preservation and/or degradation.*

L49-50 Wahrhaftig and Cox (1959) reference - references to more recent publications would seem appropriate here

Response: Thank you for your suggestions. We have included Haeberli et al (2010).

*Haeberli W., Noetzli J., Arenson L., Delaloye R., Gärtner-Roer I., Gruber S., Isaksen K., Kneisel C., Krautblatter M. and Phillips M., 2010. Mountain permafrost: development and challenges of a young research field. Journal of Glaciology, Vol. 56, No. 200.*

L50-51 "closely related to their equilibrium state" - provide reference

Response: Thank you for your suggestions. We have included the following references:

*Studies as Kääb et al (2007) and Bodin et al (2009) suggest that the increases in rock glacier velocities are a consequence of higher surface temperature.*

*Kääb A, Frauenfelder R, Roer I. 2007. On the response of rockglacier creep to surface temperature increase. Global and Planetary Change 56: 172–187. DOI. 10.1016/j.gloplacha.2006.07.005*

*Bodin X, Thibert E, Fabre D, Ribolini A, Schoeneich P, Francou B, Reynaud L, Fort M. 2009. Two decades of responses (1986–2006) to climate by the Laurichard rock glacier, French Alps. Permafrost and Periglacial Processes 20: 331–344. DOI. 10.1002/ppp.665*

L56 That depends on the type of inactivity; see e.g. Barsch (1996)

Response: Thank you for your suggestions. We have clarified the sentence. In our view, Barsch 1996 mentioned two types of inactive rock glaciers: climatic or dynamic. In the first case inactivity is due to the ice melting and in the second case the inactivity is because rock glaciers surface had moves away from the debris source. Rock glaciers in the study area; in general have a small size with frontal talus not very far from the contribution zone. This is why we assume that the activity is mainly due to climatic factors.

*Barsch, D.: Rockglaciers. Indicators for the Present and Former Geoecology in High Mountain Environments, Vol. 16, Springer, Berlin, 1996*

L58 protalus rampart - If they can be considered small rock glaciers, why use the concept of protalus rampart? Can protalus ramparts be active, inactive and relict?

Response: Thank you for your suggestions. Protalus ramparts have not been considered as small rock glaciers, they are considered as an expression of mountain permafrost creep, as had defined Barsch (1996) and more recently Scapozza et al (2011) and Hedding (2011).

*Scapozza, C., C. Lambiel, L. Baron, L. Marescot and E. Reynard 2011. Internal structure and permafrost distribution in two alpine periglacial talus slopes, Valais, Swiss Alps. Geomorphology, 132(3–4), 208–221.*

*Hedding, D. W. 2011. Pronival rampart and protalus rampart: A review of terminology. Journal of Glaciology 57(206):1179-1180*

L89 To my knowledge, this area is located south of the Arid Diagonal. Please refer to the climatological literature.

Response: Thank you for your suggestions. We propose as an alternative:

*The climate of this region is clearly influenced by the South American Arid Diagonal (Font and Chiesa, 2015)*

*Font E. and Chiesa J., 2015. Palaeoenvironmental reconstruction based on charophytes and sedimentology: Can the mid-Holocene Optimum be recognized in western Argentina?. Aquatic Botany 120 (2015) 31–38*

L90 Aridity in this region is not primarily due to the "rain shadow" of the Andes. Please refer to the climatological literature.

Response: Thank you for your suggestions. We have included Rivera et al., 2012.

*Rivera J., Penalbab C. and Betollia L., 2012. Inter-annual and inter-decadal variability of dry days in Argentina. International Journal of Climatology 33: 834–842 (2013). Published online 20 March 2012 in Wiley Online Library (wileyonlinelibrary.com) DOI: 10.1002/joc.3472*

L93 Is the Zonda relevant for this study, which focuses on high-altitude regions in the central part of the Andes, not the valleys and foreland toward the East?

Response: Thank you for your suggestions. We have included this information in the revised version of the manuscript

L95 "close" relationship with ENSO is overstated

Response: Thank you for your suggestions. We have incorporated the relevant information in the revised paper. Leiva (1999) and Masiokas (2006) show that when occurred warm El Niño-Southern Oscillation (ENSO) phenomena, there are a noticeable reduction over glacier and snowpack.

L97-100 This summary of the relationship between ENSO and glacier mass balance is

weak and of little relevance for the present study

Response: Thank you for your suggestions. We have incorporated the relevant information in the revised paper.

L101-103 MAAT for which years? are these years representative?

Response: This measure was obtained by Schreiber (2015), between April 2011 and February 2014 with stations located in the study area and close to it. These are the only data measured in the area that is why we consider mentioning.

*Schreiber, E.: Modeling the distribution of Mountain Permafrost in Central Andes, San Juan, Argentina, M. S. thesis, University of Delaware, United States, 2015*

L104-106 This section overstates the degree to which Pleistocene glaciations have modified the landscape.

Response: we believe that this is an important contribution as the mentioned glacial landforms had been observed and described.

L110 "As regards glaciers" - none of the features mentioned in this sentence is considered a glacier

Response: Thank you for your suggestions. We have included this information in the revised version of the manuscript. "As regards periglacial landforms".

L116-121 Unnecessary structural geological detail

Response: Thank you for your suggestions. We have included this information in the revised version of the manuscript. The new paragraph would be: The structural, topographic, stratigraphic and geological aspects are highly dependent on the geometry of the Wadatti Benioff zone (Smalley and Isacks, 1990), where the interaction between Southamerican and Nazca tectonics plates take place.

L154ff - Provide references supporting the relationship between form and activity status. Roer and Nyenhuis (2007) provide a comprehensive summary of criteria, which should be considered more carefully. Azócar et al. (2017) in The Cryosphere apply similar criteria to an area in the semi-arid Andes.

Response: Thank you for your suggestions. We have included this information in the revised version of the manuscript.

*- Rock glaciers are distinctive geomorphological landforms, easily identifiable through satellite images with a high spatial resolution (Wahrhaftig and Cox, 1959; Haeberli, 1985; Roer and Nyenhuis, 2007). These may extend from the rock wall, outward and downslope with a moderated slope and a frontal talus with a pronunciated slope. Above the rock glacier surface could be develop furrows and ridges or collapsed structures associated to ice melting. Intact rock glaciers were differed as active and inactive, using geomorphological criterial as the degree of furrows and ridges development, among others. Active rock glaciers present a better development, also these are characterized by exposure of fine debris at the talus front and large block accumulation at the talus bottom. Other geomorphological criteria used was the angle of the frontal talus slope, which is major on active rock glacier. The existence of vegetation, collapsed structures and flat surfaces are characteristics of inactive rock glaciers. In the Figures 1 and 2 there are two examples of active and inactive rock glaciers mapping.*

*- Fossil or relict rock glaciers have lost all their ice content and indicate current environmental conditions unfavorable for their development. They have the same shape as rock glaciers but their surfaces are completely devoid of periglacial processes, and looks completely depressed. Vegetation development on its surface is common, and the frontal talus is usually underdeveloped or even could been disappeared.*

*- Protalus Rampart have been considered an expression of mountain permafrost creep and, as such, they may be considered as embryonic rock glaciers (Barsch, 1996). Therefore the criteria to distinguish them is their dimension, these are smaller than a rock glacier. Generally the width is greater than the length and their surface, practically, does not have furrows and ridges development.*

*For each mapped landform, was considered as the upper limit the root zone where is produced the abrupt slope change between the rock wall and the rock glacier or protalus rampart surface. While, for the lower limit, was included the lower part of the frontal and lateral talus of the rock glaciers or protalus rampart.*

[Figure]

Fig1: a- Active rock glacier mapping. Geographic coordinates: 31° 9' 45" S and 70° 12' 50" W

b- Field Control. Furrows and ridges over rock glacier surface. Two people as scale.

[Figure]

Fig. 2: a- Inactive rock glacier mapping. Geographic coordinates: 31° 2'45"S 70°15'50"W. b- Field Control. Low frontal talus slope and a depressed general structure.

L154 "thermodynamic equilibrium" - is it appriopriate to apply this concept here, in particular to inactive rock glaciers that may be degrading?

Response: Thank you for your suggestions. We have corrected as follows *"equilibrium with environment"*

L192 provide reference(s); is this a state-of-the-art technique? The frequency-ratio method requires splitting of predictors into discrete classes. These classes are poorly justified in this study. Methods such as logistic regression or generalized additive models allow for continuous relationships and are capable of accounting for the influence of each predictor while accounting for the other predictors.

Response: Thank you for your suggestions. For the frequency ratio method, the predictors were splitting into justified discrete classes. For this purpose, we have considered the distribution of rock glaciers over the analyzed parameters.

We have included another method, such as logistic regression to accounting for the influence of each predictor.*(see Appendix 1)

L199 The goodness-of-fit of the model should be assessed using criteria that are to be indicated in this section.

Response: Thank you for your suggestion about describe the method assessed with detail.

The goodness of fit of our statistical model are based on Hosmer and Lemeshow Test (Table 1) and Pearson's chi-squared test (see table 3 in Appendix 1). The test showed that the goodness of fit of the equation could be accepted because the values of Cox and Snell $R^2$ (0.584) and Nagelkerke $R^2$ (0.779) are greater than 0.2 (Clark and Hosking, 1986).

Table 1: Hosmer and Lemeshow Test (H-L Test)

| Step | Chi squared | gl | Sig |
|------|-------------|-----|-------|
| 1 | 7,678 | 8 | 0,466 |

L204-206 This paragraph does not report results.

Response: Thank you for your suggestions. We have corrected in the revise version of the manuscript. The results are presented mainly in Figure 4 cited in this paragraph.

L209-210 How do the authors know that seasonal frost is of particular importance in this elevation range? Rock glaciers and their possible preservation or degradation are not primarily driven by seasonal temperature fluctuations.

Response: Thank you for your comment. In our view, Belt 1 is the lowest altitudinal belt (below 3.300 m.a.s.l.) and is considered as Seasonal Frost Belt because we consider that periglacial phenomena are limited to seasonal freezing and thawing. This not exclude that this phenomenon occurs over higher zones than belt 1.

L222ff The authors need to distinguish between the distribution of the environmental variables conditional on rock glacier presence (or activity status), and the occurrence of rock glaciers (or their activity status) conditional on these variables. The former way of conditioning does not account for the different frequencies (in the landscape) of, e.g., west-facing and south-facing slopes. Again, a multiple-variable logistic regression model would account for possible confounders.

Response: Thank you for your suggestions. We have included this information in the revised version of the manuscript. To facilitate reading we separate the values, as suggested by Referee #1 and Referee #2. Also we have added a multiple-variable logistic regression, as suggested by referee 2. *(see Appendix 1)

 The mean slope in active rock glaciers has minimum values of 11,2%, maxims of 28,5% and averages of 18,9%. The mean slope in inactive rock glaciers has minimum values of 8,5%, maxims of 26% and averages of 18,5%. The mean slope in fossil rock glaciers has minimum values of 3,9%, maxims of 31,3% and averages of 12,7% and protalus ramparts mean slope has minimum values of 4,5%, maxims of 48,8% and averages of 4,5%. This shows that a very high slope (greater than 30%) is not related to rock glacier occurrence (Fig. 6).

L247-248 The mentioned Oligocene class represents only 0.57% of the study area; its frequency ratio has insufficient empirical support.

Thank you for your suggestions. We have included this information in the discussion section, comparing with logistic regression results.

L255-256 The empirical relationship between lithology (and effectively only one of its classes) and rock glacier occurrence can be confounded with other possible factors such as slope angle or solar radiation since the frequency ratio method is unable to separate these empirical relationships. Furthermore, a possible causal link related to specific characteristics of these rocks should be established before claiming that periglacial landforms (or specifically, rock glaciers) are "affected" by lithology.

Response: Thank you for your suggestions. We have included this information in the *( Appendix 1)

L276 Much of the material covered in the Conclusions section should be part of the Discussion section where it should be discussed in the context of the available literature; part of this section also simply repeats some of the results and interpretations made before. The current Discussion (considering both sections 5 and 6) of results, their interpretation, uncertainties and limitations of methodology, is currently too weak.

Response: Thank you for your suggestions. We have included this information in the revise version of the manuscript.

**Logistic Regression**

**Methodology**

Using the logistic regression model, relationships between the presence of rock glaciers and different variables were analyzed. Logistic regression allows the formation of multivariate regression relation between a dependent variable and several independent variables (Hosmer and Lemeshow, 1989; Atkinson and Massari, 1998). Dependent data are made up of 0 and 1 values which show the absence and presence of rock glaciers respectively. In the current situation, the binary dependent variable represents the presence or absence of an active rock glacier. The forward stepwise logistic regression was carried out to incorporate predictor variables with an important contribution to the presence of rock glaciers. In a logistic regression analysis, the number of points representing areas with a occurrence and that without it should be the same (e.g., Ayalew and Yamagishi, 2005). In the study area, 1000 points, represent the rock glaciers. Therefore, 1000 points without rock glaciers were randomly selected for logistic regression.

In logistic regression, multicollinearity checking is necessary to check the correlation of independent variables (Hosmer and Lemeshow, 1989). Multicollinearity is a statistical situation in which two or more predictor variables are highly correlated, meaning that one can be linearly predicted from the others with a non-trivial degree of accuracy. Tolerance (TOL) and the variance inflation factor are two important indexes that are widely used for multicollinearity checking. According to Menard (1995), a TOL value less than 0.2 is one indicator for multicollinearity, and serious multicollinearity occurs between independent variables when the TOL values are smaller than 0.1. The variance inflation factor (VIF) is calculated by 1/tolerance. If VIF value exceeds 10, it is often regarded as indicating multicollinearity. Additionally, the Pearson correlation was also used to test the correlation between variables.

**Result**

The observed distribution of active rock glaciers and logistic regression coefficients are shown in Table 1. These information will be added to Table 2 (In Discussion Paper), where are available the number of pixels showing active rock glaciers occurrence, pixels in domainc and percentage of pixels showing rock glaciers occurrence. With, all the factors treated as categorical variable, the system constructed were found to be valid; with 89.1% of the pixels used being correctly predicted (90.9% of the active rock glaciers pixels and 87.3% of non-rock glaciers).

The test showed that the goodness of fit of the equation could be accepted because the values of Cox and Snell R2 (0.584) and Nagelkerke R2 (0.779) are greater than 0.2 (Clark and Hosking, 1986). The TOL, VIF values and Pearson correlations in this study are showed in

Table 2 and 3. It reveals that there is no multicolinearity among any of the variables and weakly correlated with each other. Pearson correlations (Table 3) show that the variables used in the present study are only weakly correlated with each other.

TABLE 1: Logistic regression coefficients

| Factor | Class | Frequency ratio | Logistic Regression |
|---|---|---|---|
| Elevation [m.a.s.l.] | 2950-3299 | 0 | -19,966 |
| | 3300-3689 | 0 | -0,748 |
| | 3690-3867 | 0,921981333 | 1,441 |
| | 3868-4047 | 3,420266369 | 1,993 |
| | 4048-4225 | 6,236900215 | 2,328 |
| | 4226-4763 | 0 | 1 |
| Aspect [degree] | 338-22 | 0,167065381 | 17,989 |
| | 23-67 | 0,301757019 | 18,084 |
| | 68-112 | 0,758833846 | 18,545 |
| | 113-157 | 1,384282336 | 20,54 |
| | 158-202 | 2,661213934 | 21,507 |
| | 203-247 | 1,949548335 | 21,117 |
| | 248-293 | 0,649774042 | 19,284 |
| | 294-337 | 0,345961807 | 18,225 |
| Slope [percentage] | 0 - 10 | 0,683108134 | -0,423 |
| | 11 - 19 | 1,764141109 | 1,33 |
| | 20 - 35 | 0,683767824 | 1,585 |
| | 35 - 75 | 0,43076071 | 1 |
| Lithology | Pliocene-Quaternary - Sedimentary deposits | 0,126154205 | -1,534 |
| | Middle to Upper Miocene - Volcanic Rocks | 1,601833172 | -0,995 |
| | Middle to Upper Miocene - Sedimentary rocks | 0 | -19,516 |
| | Oligocene to Lower Miocene - Intrusive Rocks | 4,651440275 | -1,385 |
| | Eocene to Oligocene - Intrusive Rocks | 0 | -20,213 |
| | Jurassic to Lower Cretaceous - Sedimentary rocks | 0 | -19,28 |
| | Permian to Triassic - Intrusive rocks | 0,395834872 | -1,001 |
| | Permian to Triassic - Vulcano-sedimentary rocks | 2,674485234 | 1 |
| | Devonian to lower permian - Intrusive rocks | 0 | -19,845 |
| Potential Solar Radiation [WH/m2] | 660 - 856788 | 0 | -0,21 |
| | 856789 - 1839402 | 1,801559359 | 21,003 |
| | 1839403 - 2322270 | 1,784156832 | 21,589 |
| | 2322271 - 2598972 | 0 | 1 |

TABLE 2: Collinearity Statistics

| Model | Collinearity Statistics | |
| --- | --- | --- |
| | Tolerance | FIV: variance inflation factor |
| Lithology | 0,858 | 1,166 |
| Slope | 0,951 | 1,052 |
| Aspect | 0,997 | 1,003 |
| Elevation | 0,807 | 1,239 |
| Potential Solar Radiation | 0,924 | 1,083 |

TABLE 3: Bilateral Correlation

| | | Lithology | Slope | Aspect | Elevation | Potential Solar Radiation |
| --- | --- | --- | --- | --- | --- | --- |
| Lithology | Pearson Correlation | 1 | 0,11 | 0,41 | 0,368 | 0,158 |
| | SIG (Bilateral) | | 0 | 0,064 | 0 | 0 |
| | N | 2000 | 2000 | 2000 | 2000 | 2000 |
| Slope | Pearson Correlation | 0,11 | 1 | 0,028 | 0,199 | 0,135 |
| | SIG (Bilateral) | 0 | | 0,205 | 0 | 0 |
| | N | 2000 | 2000 | 2000 | 2000 | 2000 |
| Aspect | Pearson Correlation | 0,041 | 0,028 | 1 | 0,021 | -0,01 |
| | SIG (Bilateral) | 0 | 0,205 | | 0,338 | 0,654 |
| | N | 2000 | 2000 | 2000 | 2000 | 2000 |
| Elevation | Pearson Correlation | 0,368 | 0,199 | 0,021 | 1 | 0,253 |
| | SIG (Bilateral) | 0 | 0 | 0,338 | | 0 |
| | N | 2000 | 2000 | 2000 | 2000 | 2000 |
| Potential Solar Radiation | Pearson Correlation | 0,158 | 0,135 | -0,01 | 0,253 | 1 |
| | SIG (Bilateral) | 0 | 0 | 0,654 | 0 | |
| | N | 2000 | 2000 | 2000 | 2000 | 2000 |

The correlation is significant at the 0.01 level (bilateral)

**References**

- Atkinson, P.M., Massari, R., 1998. Generalised linear modelling of susceptibility to landsliding in the central Appenines, Italy. Computers and Geosciences 24 (4),373-385.

- Hosmer, D.W., Lemeshow, S., 1989. Applied Logistic Regression. John Wiley & Sons, Inc, New York, p. 307.

- Ayalew L, Yamagishi H, 2005. The application of GIS-based logistic regression for landslide susceptibility mapping in the Kakuda-Yahiko Mountains, Central Japan. Geomorphology 65, 15–31.

- Menard, S.W. ,1995. Applied Logistic Regression Analysis. SAGE, Thousand Oaks, p. 128.

NASA., 2011. ASTER Global Digital Elevation Map V2. http://gdem.ersdac. jspacesystems.or.jp.

- Clark, W.A., Hosking, P.L., 1986. Statistical Methods for Geographers. Wiley, New York.